# GRADIENT DESCENT TEMPORAL DIFFERENCE-DIFFERENCE LEARNING

## ABSTRACT

Off-policy algorithms, in which a behavior policy differs from the target policy and is used to gain experience for learning, have proven to be of great practical value in reinforcement learning. However, even for simple convex problems such as linear value function approximation, these algorithms are not guaranteed to be stable. To address this, alternative algorithms that are provably convergent in such cases have been introduced, the most well known being gradient descent temporal difference (GTD) learning. This algorithm and others like it, however, tend to converge much more slowly than conventional temporal difference learning. In this paper we propose *gradient descent temporal difference-difference (Gradient-DD) learning* in order to improve GTD learning by introducing second-order differences in successive parameter updates. We investigate this algorithm in the framework of linear value function approximation, analytically showing its improvement over GTD learning. Studying the model empirically on the random walk and Boyan-chain prediction tasks, we find substantial improvement over GTD learning and, in several cases, better performance even than conventional TD learning.

## 1 INTRODUCTION

Off-policy algorithms for value function learning enable an agent to use a behavior policy that differs from the target policy in order to gain experience for learning. However, because off-policy methods learn a value function for a target policy given data due to a different behavior policy, they often exhibit greater variance in parameter updates. When applied to problems involving function approximation, off-policy methods are slower to converge than on-policy methods and may even diverge (Baird, 1995; Sutton & Barto, 2018).

Two general approaches have been investigated to address the challenge of developing stable and effective off-policy temporal-difference algorithms. One approach is to use importance sampling methods to warp the update distribution back to the on-policy distribution (Precup et al., 2000; Mahmood et al., 2014). This approach is useful for decreasing the variance of parameter updates, but it does not address stability issues. The second main approach to addressing the challenge of off-policy learning is to develop true gradient descent-based methods that are guaranteed to be stable regardless of the update distribution. Sutton et al. (2009a;b) proposed the first off-policy gradient-descent-based temporal difference (GTD and GTD2, respectively) algorithms. These algorithms are guaranteed to be stable, with computational complexity scaling linearly with the size of the function approximator. Empirically, however, their convergence is much slower than conventional temporal difference (TD) learning, limiting their practical utility (Ghiassian et al., 2020; White & White, 2016). Building on this work, extensions to the GTD family of algorithms (see Ghiassian et al. (2018) for a review) have allowed for incorporating eligibility traces (Maei & Sutton, 2010; Geist & Scherrer, 2014), non-linear function approximation such as with a neural network (Maei, 2011), and reformulation of the optimization as a saddle point problem (Liu et al., 2015; Du et al., 2017). However, due to their slow convergence, none of these stable off-policy methods are commonly used in practice.

In this work, we introduce a new gradient descent algorithm for temporal difference learning with linear value function approximation. This algorithm, which we call *gradient descent temporal difference-difference* (Gradient-DD) learning, is an acceleration technique that employs second-

order differences in successive parameter updates. The basic idea of Gradient-DD is to modify the error objective function by additionally considering the prediction error obtained in last time step, then to derive a gradient-descent algorithm based on this modified objective function. In addition to exploiting the Bellman equation to get the solution, this modified error objective function avoids drastic changes in the value function estimate by encouraging local search around the current estimate. Algorithmically, the Gradient-DD approach only adds an additional term to the update rule of the GTD2 method, and the extra computational cost is negligible. We show mathematically that applying this method significantly improves the convergence rate relative to the GTD2 method for linear function approximation. This result is supported by numerical experiments, which also show that Gradient-DD obtains better convergence in many cases than conventional TD learning.

## 1.1 RELATED WORK

In related approaches to ours, some previous studies have attempted to improve Gradient-TD algorithms by adding regularization terms to the objective function. Liu et al. (2012) have used $l_1$ regularization on weights to learn sparse representations of value functions, and Ghiassian et al. (2020) has used $l_2$ regularization on weights. Unlike these references, our approach modifies the error objective function by regularizing the evaluation error obtained in the most recent time step. With this modification, our method provides a learning rule that contains second-order differences in successive parameter updates.

Our approach is similar to trust region policy optimization (Peters & Schaal, 2008; Schulman et al., 2015) or relative entropy policy search (Peters et al., 2010), which penalize large changes being learned in policy learning. In these methods, constrained optimization is used to update the policy by considering the constraint on some measure between the new policy and the old policy. Here, however, our aim here is to look for the optimal value function, and the regularization term uses the previous value function estimate to avoid drastic changes in the updating process.

## 2 GRADIENT DESCENT METHOD FOR OFF-POLICY TEMPORAL DIFFERENCE LEARNING

### 2.1 PROBLEM DEFINITION AND BACKGROUND

In this section, we formalize the problem of learning the value function for a given policy under the Markov Decision Process (MDP) framework. In this framework, the agent interacts with the environment over a sequence of discrete time steps, $t = 1, 2, \ldots$. At each time step the agent observes a partial summary of the state $s_t \in \mathcal{S}$ and selects an action $a_t \in \mathcal{A}$. In response, the environment emits a reward $r_t \in \mathbb{R}$ and transitions the agent to its next state $s_{t+1} \in \mathcal{S}$. The state and action sets are finite. State transitions are stochastic and dependent on the immediately preceding state and action. Rewards are stochastic and dependent on the preceding state and action, as well as on the next state. The process generating the agent's actions is termed the behavior policy. In off-policy learning, this behavior policy is in general different from the target policy $\pi : \mathcal{S} \to \mathcal{A}$. The objective is to learn an approximation to the state-value function under the target policy in a particular environment:

$$V(s) = \mathrm{E}_\pi \left[ \sum_{t=1}^{\infty} \gamma^{t-1} r_t | s_1 = s \right],$$ (1)

where $\gamma \in [0, 1)$ is the discount rate.

In problems for which the state space is large, it is practical to approximate the value function. In this paper we consider linear function approximation, where states are mapped to feature vectors with fewer components than the number of states. Specifically, for each state $s \in \mathcal{S}$ there is a corresponding feature vector $\mathbf{x}(s) \in \mathbb{R}^p$, with $p \leq |\mathcal{S}|$, such that the approximate value function is given by

$$V_{\mathbf{w}}(s) := \mathbf{w}^\top \mathbf{x}(s).$$ (2)

The goal is then to learn the parameters $\mathbf{w}$ such that $V_{\mathbf{w}}(s) \approx V(s)$.

## 2.2 Gradient temporal difference learning

A major breakthrough for the study of the convergence properties of MDP systems came with the introduction of the GTD and GTD2 learning algorithms (Sutton et al., 2009a;b). We begin by briefly recapitulating the GTD algorithms, which we will then extend in the following sections. To begin, we introduce the Bellman operator $\mathbf{B}$ such that the true value function $\mathbf{V} \in \mathbb{R}^{|S|}$ satisfies the Bellman equation:

$$\mathbf{V} = \mathbf{R} + \gamma \mathbf{P} \mathbf{V} =: \mathbf{B} \mathbf{V},$$

where $\mathbf{R}$ is the reward vector with components $\mathrm{E}(r_{n+1}|s_n = s)$, and $\mathbf{P}$ is a matrix of state transition probabilities. In temporal difference methods, an appropriate objective function should minimize the difference between the approximate value function and the solution to the Bellman equation.

Having defined the Bellman operator, we next introduce the projection operator $\mathbf{\Pi}$, which takes any value function $\mathbf{V}$ and projects it to the nearest value function within the space of approximate value functions of the form (2). Letting $\mathbf{X}$ be the matrix whose rows are $\mathbf{x}(s)$, the approximate value function can be expressed as $\mathbf{V_w} = \mathbf{X} \mathbf{w}$. We will also assume that there exists a limiting probability distribution such that $d_s = \lim_{n \to \infty} p(s_n = s)$ (or, in the episodic case, $d_s$ is the proportion of time steps spent in state $s$). The projection operator is then given by

$$\mathbf{\Pi} = \mathbf{X}(\mathbf{X}^\top \mathbf{D} \mathbf{X})^{-1} \mathbf{X}^\top \mathbf{D},$$

where the matrix $\mathbf{D}$ is diagonal, with diagonal elements $d_s$.

The natural measure of how closely the approximation $\mathbf{V_w}$ satisfies the Bellman equation is the mean-squared Bellman error:

$$\mathrm{MSBE}(\mathbf{w}) = \|\mathbf{V_w} - \mathbf{B}\mathbf{V_w}\|_{\mathbf{D}}^2, \tag{3}$$

where the norm is weighted by $\mathbf{D}$, such that $\|\mathbf{V}\|_{\mathbf{D}}^2 = \mathbf{V}^\top \mathbf{D} \mathbf{V}$. However, because the Bellman operator follows the underlying state dynamics of the Markov chain, irrespective of the structure of the linear function approximator, $\mathbf{B}\mathbf{V_w}$ will typically not be representable as $\mathbf{V_w}$ for any $\mathbf{w}$. An alternative objective function, therefore, is the mean squared *projected* Bellman error (MSPBE), which we define as

$$J(\mathbf{w}) = \|\mathbf{V_w} - \mathbf{\Pi}\mathbf{B}\mathbf{V_w}\|_{\mathbf{D}}^2. \tag{4}$$

Following (Sutton et al., 2009b), our objective is to minimize this error measure. As usual in stochastic gradient descent, the weights at each time step are then updated by $\Delta \mathbf{w} = -\alpha \nabla_{\mathbf{w}} J(\mathbf{w})$, where $\alpha > 0$, and

$$-\frac{1}{2} \nabla_{\mathbf{w}} J(\mathbf{w}) = - \mathrm{E}[(\gamma \mathbf{x}_{n+1} - \mathbf{x}_n)\mathbf{x}_n^\top][\mathrm{E}(\mathbf{x}_n \mathbf{x}_n^\top)]^{-1} \mathrm{E}(\delta_n \mathbf{x}_n)$$

$$\approx - \mathrm{E}[(\gamma \mathbf{x}_{n+1} - \mathbf{x}_n)\mathbf{x}_n^\top] \boldsymbol{\eta}. \tag{5}$$

For notational simplicity, we have denoted the feature vector associated with $s_n$ as $\mathbf{x}_n = \mathbf{x}(s_n)$. We have also introduced the temporal difference error $\delta_n = r_n + (\gamma \mathbf{x}_{n+1} - \mathbf{x}_n)^\top \mathbf{w}_n$, as well as $\boldsymbol{\eta}$, a linear predictor to approximate $[\mathrm{E}(\mathbf{x}_n \mathbf{x}_n^\top)]^{-1} \mathrm{E}(\delta_n \mathbf{x}_n)$. Because the factors in Eqn. (5) can be directly sampled, the resulting updates in each step are

$$\delta_n = r_n + (\gamma \mathbf{x}_{n+1} - \mathbf{x}_n)^\top \mathbf{w}_n$$
$$\boldsymbol{\eta}_{n+1} = \boldsymbol{\eta}_n + \beta_n(\delta_n - \mathbf{x}_n^\top \boldsymbol{\eta}_n)\mathbf{x}_n$$
$$\mathbf{w}_{n+1} = \mathbf{w}_n - \alpha_n(\gamma \mathbf{x}_{n+1} - \mathbf{x}_n)(\mathbf{x}_n^\top \boldsymbol{\eta}_n). \tag{6}$$

These updates define the GTD2 learning algorithm, which we will build upon in the following section.

## 3 Gradient descent temporal difference-difference learning

In order to improve the GTD2 algorithm described above, in this section we modify the objective function via additionally considering the approximation error $\mathbf{V_w} - \mathbf{V}_{\mathbf{w}_{n-1}}$ given the previous time step $n-1$. Specifically, we modify Eqn. (4) as follows:

$$J_{\mathrm{GDD}}(\mathbf{w}|\mathbf{w}_{n-1}) = J(\mathbf{w}) + \kappa \|\mathbf{V_w} - \mathbf{V}_{\mathbf{w}_{n-1}}\|_{\mathbf{D}}^2, \tag{7}$$

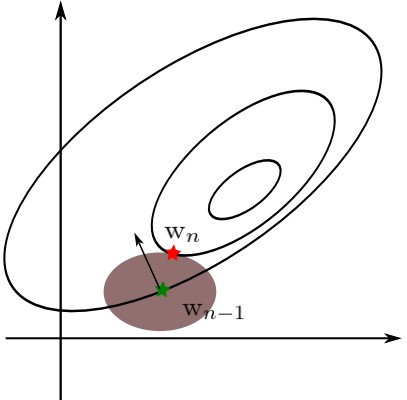

Figure 1: Schematic diagram of Gradient-DD learning with $\mathbf{w} \in \mathbb{R}^2$. Rather than updating $\mathbf{w}$ directly along the gradient of the MSPBE (arrow), the update rule selects $\mathbf{w}_n$ that minimizes the MSPBE while satisfying the constraint $\|\mathbf{V_w} - \mathbf{V}_{\mathbf{w}_{n-1}}\|_{\mathbf{D}}^2 \leq \mu$ (shaded ellipse).

where $\kappa \geq 0$ is a parameter of the regularization.

Minimizing Eqn. (7) is equivalent to the following optimization

$$\arg \min_{\mathbf{w}} J(\mathbf{w}) \text{ s.t. } \|\mathbf{V_w} - \mathbf{V}_{\mathbf{w}_{n-1}}\|_{\mathbf{D}}^2 \leq \mu \tag{8}$$

where $\mu > 0$ is a parameter which becomes large when $\kappa$ is small, so that the MSPBE objective is recovered as $\mu \to \infty$, equivalent to $\kappa \to 0$ in Eqn. (7). We show in the Appendix that for any $\mu > 0$, there exist $\kappa \geq 0$ such that the solution of Eqn. (7) and that of Eqn. (8) are the same.

Eqns. (7) and (8) represent a tradeoff between minimizing the MSPBE error and preventing the estimated value function from changing too drastically. Rather than simply minimizing the optimal prediction from the projected Bellman equation, the agent makes use of the most recent update to look for the solution. Figure 1 gives a schematic view of the effect of the regularization. Rather than directly following the direction of the MSPBE gradient, the update chooses a $\mathbf{w}$ that minimizes the MSPBE while following the constraint that the estimated value function should not change too greatly. In effect, the regularization term encourages searching around the estimate at previous time step, especially when the state space is large.

With these considerations in mind, the negative gradient of $J_{\text{GDD}}(\mathbf{w}|\mathbf{w}_{n-1})$ is

$$
\begin{aligned}
&-\frac{1}{2}\nabla_{\mathbf{w}} J_{\text{GDD}}(\mathbf{w}|\mathbf{w}_{n-1}) \\
={}&- \mathrm{E}[(\gamma\mathbf{x}_{n+1} - \mathbf{x}_n)\mathbf{x}_n^\top][\mathrm{E}(\mathbf{x}_n\mathbf{x}_n^\top)]^{-1}\mathrm{E}(\delta_n\mathbf{x}_n) - \kappa\mathrm{E}[(\mathbf{x}_n^\top\mathbf{w}_n - \mathbf{x}_n^\top\mathbf{w}_{n-1})\mathbf{x}_n] \\
\approx{}&- \mathrm{E}[(\gamma\mathbf{x}_{n+1} - \mathbf{x}_n)\mathbf{x}_n^\top]\boldsymbol{\eta}_n - \kappa\mathrm{E}[(\mathbf{x}_n^\top\mathbf{w}_n - \mathbf{x}_n^\top\mathbf{w}_{n-1})\mathbf{x}_n].
\end{aligned}
\tag{9}
$$

Because the terms in Eqn. (9) can be directly sampled, the stochastic gradient descent updates are given by

$$
\begin{aligned}
\delta_n ={}& r_n + (\gamma\mathbf{x}_{n+1} - \mathbf{x}_n)^\top\mathbf{w}_n \\
\boldsymbol{\eta}_{n+1} ={}& \boldsymbol{\eta}_n + \beta_n(\delta_n - \mathbf{x}_n^\top\boldsymbol{\eta}_n)\mathbf{x}_n \\
\mathbf{w}_{n+1} ={}& \mathbf{w}_n - \kappa_n(\mathbf{x}_n^\top\mathbf{w}_n - \mathbf{x}_n^\top\mathbf{w}_{n-1})\mathbf{x}_n - \alpha_n(\gamma\mathbf{x}_{n+1} - \mathbf{x}_n)(\mathbf{x}_n^\top\boldsymbol{\eta}_n).
\end{aligned}
\tag{10}
$$

These update equations define the Gradient-DD method, in which the GTD2 update equations (6) are generalized by including a second-order update term in the third update equation, where this term originates from the squared bias term in the objective (7). In the following sections, we shall analytically and numerically investigate the convergence and performance of Gradient-DD learning.

## 4  IMPROVED CONVERGENCE RATE

In this section we analyze the convergence rate of Gradient-DD learning. Note that the second-order update in the last line in Eqn. (10) can be rewritten as a system of first-order difference equations:

$$
\begin{aligned}
(\mathbf{I} + \kappa_n\mathbf{x}_n\mathbf{x}_n^\top)(\mathbf{w}_{n+1} - \mathbf{w}_n) ={}& \kappa_n\mathbf{x}_n\mathbf{x}_n^\top(\mathbf{u}_{n+1} - \mathbf{u}_n) - \alpha_n(\gamma\mathbf{x}_{n+1} - \mathbf{x}_n)(\mathbf{x}_n^\top\boldsymbol{\eta}_n); \\
\mathbf{u}_{n+1} ={}& \mathbf{w}_{n+1} - \mathbf{w}_n.
\end{aligned}
\tag{11}
$$

Let $\beta_n = \zeta\alpha_n$, $\zeta > 0$. We consider constant step sizes in the updates, i.e., $\kappa_n = \kappa$ and $\alpha_n = \alpha$. Denote $\mathbf{H}_n = \begin{bmatrix} \mathbf{0} & \mathbf{0} \\ \mathbf{0} & \mathbf{x}_n\mathbf{x}_n^\top \end{bmatrix}$ and $\mathbf{G}_n = \begin{bmatrix} \sqrt{\zeta}\mathbf{x}_n\mathbf{x}_n^\top & \mathbf{x}_n(\mathbf{x}_n - \gamma\mathbf{x}_{n+1})^\top \\ -(\mathbf{x}_n - \gamma\mathbf{x}_{n+1})\mathbf{x}_n^\top & \mathbf{0} \end{bmatrix}$. We rewrite the update rules of two iterations in Eqn. (11) as a single iteration in a combined parameter vector with $2n$ components, $\boldsymbol{\rho}_n = (\boldsymbol{\eta}_n^\top/\sqrt{\zeta}, \mathbf{w}_n^\top)^\top$, and a new reward-related vector with $2n$ components, $\mathbf{g}_{n+1} = (r_n\mathbf{x}_n^\top, \mathbf{0}^\top)^\top$, as follows:

$$\boldsymbol{\rho}_{n+1} = \boldsymbol{\rho}_n - \kappa\mathbf{H}_n(\boldsymbol{\rho}_n - \boldsymbol{\rho}_{n-1}) + \sqrt{\zeta}\alpha(\mathbf{G}_n\boldsymbol{\rho}_n + \mathbf{g}_{n+1}), \qquad (12)$$

Denoting $\boldsymbol{\psi}_{n+1} = \alpha^{-1}(\boldsymbol{\rho}_{n+1} - \boldsymbol{\rho}_n)$, Eqn. (12) is rewritten as

$$
\begin{aligned}
\begin{bmatrix} \boldsymbol{\rho}_{n+1} - \boldsymbol{\rho}_n \\ \boldsymbol{\psi}_{n+1} - \boldsymbol{\psi}_n \end{bmatrix} &= \alpha \begin{bmatrix} \mathbf{I} + \kappa\mathbf{H}_n & -\kappa\alpha\mathbf{H}_n \\ \mathbf{I} & -\alpha\mathbf{I} \end{bmatrix}^{-1} \begin{bmatrix} -\sqrt{\zeta}(\mathbf{G}_n\boldsymbol{\rho}_n - \mathbf{g}_{n+1}) \\ \boldsymbol{\psi}_n \end{bmatrix} \\
&= \alpha \begin{bmatrix} -\sqrt{\zeta}\mathbf{G}_n & -\kappa\mathbf{H}_n \\ -\sqrt{\zeta}\alpha^{-1}\mathbf{G}_n & -\alpha^{-1}(\mathbf{I}+\kappa\mathbf{H}_n) \end{bmatrix} \begin{bmatrix} \boldsymbol{\rho}_n \\ \boldsymbol{\psi}_n \end{bmatrix} + \alpha \begin{bmatrix} \sqrt{\zeta}\mathbf{g}_{n+1} \\ \sqrt{\zeta}\alpha^{-1}\mathbf{g}_{n+1} \end{bmatrix},
\end{aligned}
$$
$$(13)$$

where the second step is from $\begin{bmatrix} \mathbf{I} + \kappa\mathbf{H}_n & -\kappa\alpha\mathbf{H}_n \\ \mathbf{I} & -\alpha\mathbf{I} \end{bmatrix}^{-1} = \begin{bmatrix} \mathbf{I} & -\kappa\mathbf{H}_n \\ \alpha^{-1}\mathbf{I} & -\alpha^{-1}(\mathbf{I}+\kappa\mathbf{H}_n) \end{bmatrix}$. Denote $\mathbf{J}_n = \begin{bmatrix} -\sqrt{\zeta}\mathbf{G}_n & -\kappa\mathbf{H}_n \\ -\sqrt{\zeta}\alpha^{-1}\mathbf{G}_n & -\alpha^{-1}(\mathbf{I}+\kappa\mathbf{H}_n) \end{bmatrix}$. Eqn. (13) tells us that $\mathbf{J}_n$ is the update matrix of the Gradient-DD algorithm. (Note that $\mathbf{G}_n$ is the update matrix of the GTD2 algorithm.) Therefore, assuming the stochastic approximation in Eqn. (13) goes to the solution of an associated ordinary differential equation (ODE) under some regularity conditions (a convergence property is provided in the appendix by following Borkar & Meyn (2000)), we can analyze the improved convergence rate of Gradient-DD learning by comparing the eigenvalues of the matrices $\mathrm{E}(\mathbf{G}_n)$ denoted by $\mathbf{G}$, and $\mathrm{E}(\mathbf{J}_n)$ denoted by $\mathbf{J}$ (Atkinson et al., 2008). Obviously, $\mathbf{J} = \begin{bmatrix} -\sqrt{\zeta}\mathbf{G} & -\kappa\mathbf{H} \\ -\sqrt{\zeta}\alpha^{-1}\mathbf{G} & -\alpha^{-1}(\mathbf{I}+\kappa\mathbf{H}) \end{bmatrix}$, where $\mathbf{H} = \mathrm{E}(\mathbf{H}_n)$. To simplify, we consider the case that the matrix $\mathrm{E}(\mathbf{x}_n\mathbf{x}_n^\top) = \mathbf{I}$.

Let $\lambda_G$ be a real eigenvalue of the matrix $\sqrt{\zeta}\mathbf{G}$. (Note that $\mathbf{G}$ is defined here with opposite sign relative to $\mathbf{G}$ in Maei (2011).) From Maei (2011), the eigenvalues of the matrix $-\mathbf{G}$ are strictly negative. In other words, $\lambda_G > 0$. Let $\lambda$ be an eigenvalue of the matrix $\mathbf{J}$, i.e. a solution to the equation

$$|\lambda\mathbf{I} - \mathbf{J}| = (\lambda + \lambda_G)(\lambda + \alpha^{-1}) + \kappa\alpha^{-1}\lambda = \lambda^2 + [\alpha^{-1}(1+\kappa) + \lambda_G]\lambda + \alpha^{-1}\lambda_G = 0. \qquad (14)$$

The smaller eigenvalues $\lambda_m$ of the pair solutions to Eqn. (14) are

$$\lambda_m < -\lambda_G,$$

where details of the above derivations are given in the appendix. This explains the enhanced speed of convergence in Gradient-DD learning. We shall illustrate this enhanced speed of convergence in numerical experiments in Section 5.

Additionally, we also show a convergence property of Gradient-DD under constant step sizes by applying the ordinary differential equation method of stochastic approximation (Borkar & Meyn, 2000). Let the TD fixed point be $\mathbf{w}^*$, such that $\mathbf{V}_{\mathbf{w}^*} = \mathbf{\Pi}\mathbf{B}\mathbf{V}_{\mathbf{w}^*}$. Under some conditions, we prove that, for any $\epsilon > 0$, there exists $b_1 < \infty$ such that $\limsup_{n\to\infty} P(\|\mathbf{w}_n - \mathbf{w}^*\| > \epsilon) \le b_1\alpha$.

Details are provided in the appendix. For tapered step sizes, which would be necessary to obtain an even stronger convergence proof, the analysis framework in Borkar & Meyn (2000) does not apply into the Gradient-DD algorithm. Although theoretical investigation of the convergence under tapered step sizes is a question to be studied, we find empirically in numerical experiments that the algorithm does in fact converge with tapered step sizes and even obtains much better performance in this case than with fixed step sizes.

## 5   EMPIRICAL STUDY

In this section, we assess the practical utility of the Gradient-DD method in numerical experiments. To validate performance of Gradient-DD learning, we compare Gradient-DD learning with GTD2

learning, TDC learning (TD with gradient correction (Sutton et al., 2009b)), TD learning, and Emphatic TD learning (Sutton & Mahmood, 2016) in tabular representation using a random-walk task and in linear representation using the Boyan-chain task. For each method and each task, we performed a scan over the step sizes $\alpha_n$ and the parameter $\kappa$ so that the comprehensive performance of the different algorithms can be compared. We considered two choices of step size sequence $\{\alpha_n\}$:

- (Case 1) $\alpha_n$ is constant, i.e., $\alpha_n = \alpha_0$.
- (Case 2) The learning rate $\alpha_n$ is tapered according to the schedule $\alpha_n = \alpha_0(10^3 + 1)/(10^3 + n)$.

We set the $\kappa = c\alpha_0$ where $c = 1, 2, 4$. Additionally, we also allow $\kappa$ dependent on $n$ and consider Case 3: $\alpha_n$ is tapered as in Case 2, but $\kappa_n = c\alpha_n$. In order to simplify presentation, the results of Case 3 are reported in the Appendix. To begin, we set $\beta_n = \alpha_n$, then later allow for $\beta_n = \zeta\alpha_n$ under $\zeta \in \{1/4, 1/2, 1, 2\}$ in order to investigate the effect of the two-timescale approach of the Gradient-based TD algorithms on Gradient-DD. In all cases, we set $\gamma = 1$.

## 5.1 RANDOM WALK TASK

As a first test of Gradient-DD learning, we conducted a simple random walk task (Sutton & Barto, 2018) with tabular representation of the value function. The random walk task has a linear arrangement of $m$ states plus an absorbing terminal state at each end. Thus there are $m + 2$ sequential states, $S_0, S_1, \cdots, S_m, S_{m+1}$, where $m = 20, 50$, or $100$. Every walk begins in the center state. At each step, the walk moves to a neighboring state, either to the right or to the left with equal probability. If either edge state ($S_0$ or $S_{m+1}$) is entered, the walk terminates. A walk's outcome is defined to be $r = 0$ at $S_0$ and $r = 1$ at $S_{m+1}$. Our aim is to learn the value of each state $V(s)$, where the true values are $(1, \cdots, m)/(m + 1)$. In all cases the approximate value function is initialized to the intermediate value $V_0(s) = 0.5$. In order to investigate the effect of the initialization $V_0(s)$, we also initialize $V_0(s) = 0$, and report the results in Figure 7 of the Appendix, where its performance is very similar as the initialization $V_0(s) = 0.5$.

We first compare the methods by plotting the empirical RMS error from the final episode during training as a function of step size $\alpha$ in Figure 2, where 5000 episodes are used. From the figure, we can make several observations. (1) Emphatic TD works well but is sensitive to $\alpha$. It prefers very small $\alpha$ even in the tapering case, and this preference becomes strong as the state space becomes large in size. (2) Gradient-DD works well and is robust to $\alpha$, as is conventional TD learning. (3) TDC performs similarly to the GTD2 method, but requires slightly larger $\alpha$ than GTD2. (4) Gradient-DD performs similarly to conventional TD learning and better than the GTD2 method. This advantage is consistent in different settings. (5) The range of $\alpha$ leading to effective learning for Gradient-DD is roughly similar to that for GTD2.

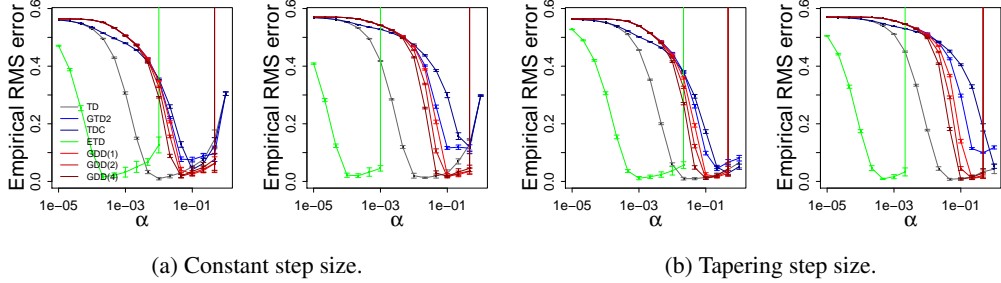

(a) Constant step size.    (b) Tapering step size.

Figure 2: Performance in the random walk task depends on step size. (a), Constant step size $\alpha_n = \alpha_0$. (b), Tapering step size $\alpha_n = \alpha_0(10^3 + 1)/(10^3 + n)$. In (a) and (b), state space size 10 (left) or 20 (right). GDD(c) denotes the Gradient-DD with $c$. The curves are averaged over 20 runs, with error bars denoting standard deviations across runs.

Next we look closely at the performance during training, which we show in Figure 3, where each method and parameter setting was run for 5000 episodes. From the observations in Figure 2, in order to facilitate comparison of these methods, we set $\alpha_0 = 0.1$ for 10 spaces, $\alpha_0 = 0.2$ for 20 spaces, and $\alpha_0 = 0.5$ for 50 spaces. Because Emphatic TD requires the step size $\alpha$ to be especially small

as shown in Figure 2, the plotted values of $\alpha_0$ for Emphatic TD are tuned relative to the values used in the algorithm defined in Sutton & Mahmood (2016), where the step sizes of Emphatic TD $\alpha_0^{(ETD)}$ are chosen from $\{0.5\%, 0.1\%, 0.05\%, 0.01\%\}$ by the smallest area under the performance curve. Additionally we also tune $\alpha_0$ for TDC because TDC requires $\alpha_n$ larger a little than GTD2 as shown in Figure 2. The step sizes for TDC are set as $\alpha_n^{(TDC)} = a\alpha_n$, where $a$ is chosen from $\{1, 1.5, 2, 3\}$ by the smallest area under the performance curve.

From the results shown in Figure 3a, we draw several observations. (1) For all conditions tested, Gradient-DD converges much more rapidly than GTD2 and TDC. The results indicate that Gradient-DD even converges faster than TD learning in some cases, though it is not as fast in the beginning episodes. (2) The advantage of Gradient-DD learning over other methods grows as the state space increases in size. (3) Gradient-DD learning is robust to the choice of $c$, which controls the size $\kappa$ of the second-order update, as long as $c$ is not too large. (Empirically $c = 2$ is a good choice.) (4) Gradient-DD has consistent and good performance under both the constant step size setting and under the tapered step size setting. In summary, compared with GTD2 learning and other methods, Gradient-DD learning in this task leads to improved learning with good convergence.

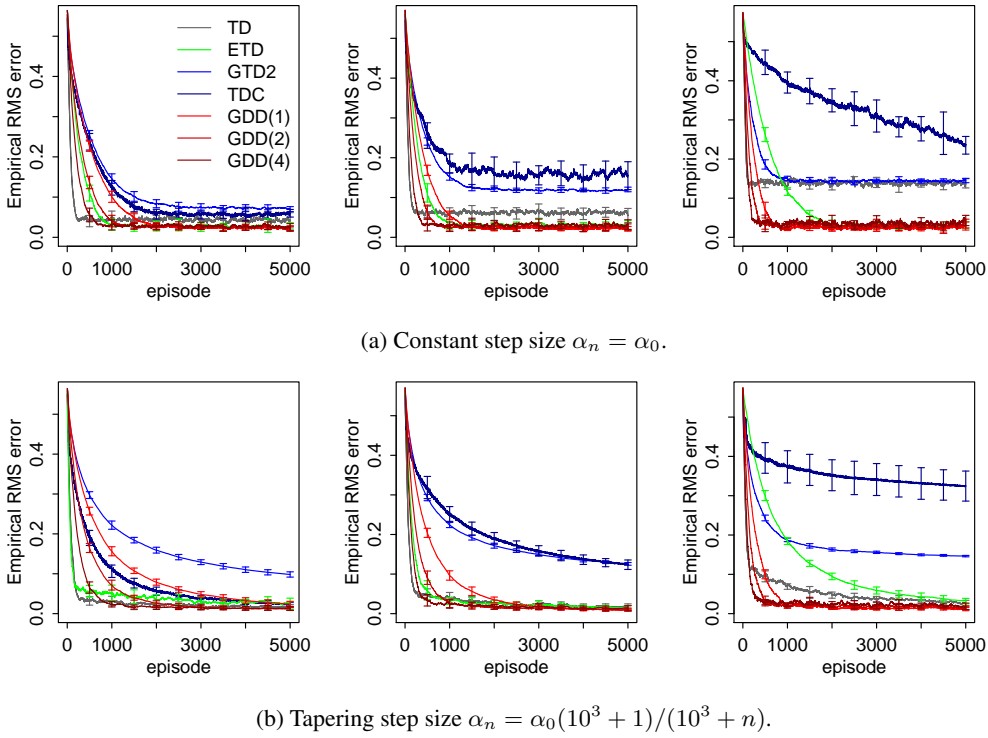

(a) Constant step size $\alpha_n = \alpha_0$.

(b) Tapering step size $\alpha_n = \alpha_0(10^3 + 1)/(10^3 + n)$.

Figure 3: Performance of Gradient-DD in the random walk task. From left to right in each subfigure: the size of state space is 10 ($\alpha_0 = 0.1$), 20 ($\alpha_0 = 0.2$), 50 ($\alpha_0 = 0.5$). The curves are averaged over 20 runs, with error bars denoting standard deviations across runs.

In addition to investigating the effects of the learning rate, size of the state space, and magnitude of the regularization parameter, we also investigated the effect of using distinct values for the two learning rates, $\alpha_n$ and $\beta_n$. To do this, we set $\beta_n = \zeta\alpha_n$ with $\zeta \in \{1/4, 1/2, 1, 2\}$ and report the results in Figure 8 of the appendix. The results show that comparably good performance of Gradient-DD is obtained under these various $\beta_n$ settings.

## 5.2 BOYAN-CHAIN TASK

We next investigate Gradient-DD learning on the Boyan-chain problem, which is a standard task for testing linear value-function approximation (Boyan, 2002). In this task we allow for $4p - 3$ states, with $p = 20$, each of which is represented by a $p$-dimensional feature vector. The $p$-dimensional representation for every fourth state from the start is $[1, 0, \cdots, 0]$ for state $s_1$, $[0, 1, 0, \cdots, 0]$ for $s_5$,

$\cdots$, and $[0, 0, \cdots, 0, 1]$ for the terminal state $s_{4p-3}$. The representations for the remaining states are obtained by linearly interpolating between these. The optimal coefficients of the feature vector are $(-4(p-1), -4(p-2), \cdots, 0)/5$. Simulations with $p = 50$ and $100$ give similar results to those from the random walk task, and hence are not shown here. In each state, except for the last one before the end, there are two possible actions: move forward one step or move forward two steps with equal probability 0.5. Both actions lead to reward -0.3. The last state before the end just has one action of moving forward to the terminal with reward -0.2. As in the random-walk task, $\alpha_0$ used in Emphatic TD is tuned from $\{0.5\%, 0.2\%, 0.1\%, 0.05\%\}$.

We report the results in Figure 4, which leads to conclusions similar to those already drawn from Figure 3. (1) Gradient-DD has much faster convergence than GTD2 and TDC, and generally converges to better values despite being somewhat slower than TD learning at the beginning episodes. (2) Gradient-DD is competitive with Emphatic TD. The improvement over other methods grows as the state space becomes larger. (3) As $\kappa$ increases, the performance of Gradient-DD improves. Additionally, the performance of Gradient-DD is robust to changes in $\kappa$ as long as $\kappa$ is not very large. Empirically a good choice is to set $\kappa = \alpha$ or $2\alpha$. (4) Comparing the performance with constant step size versus that with tapered step size, the Gradient-DD method performs better with tapered step size than it does with constant step size.

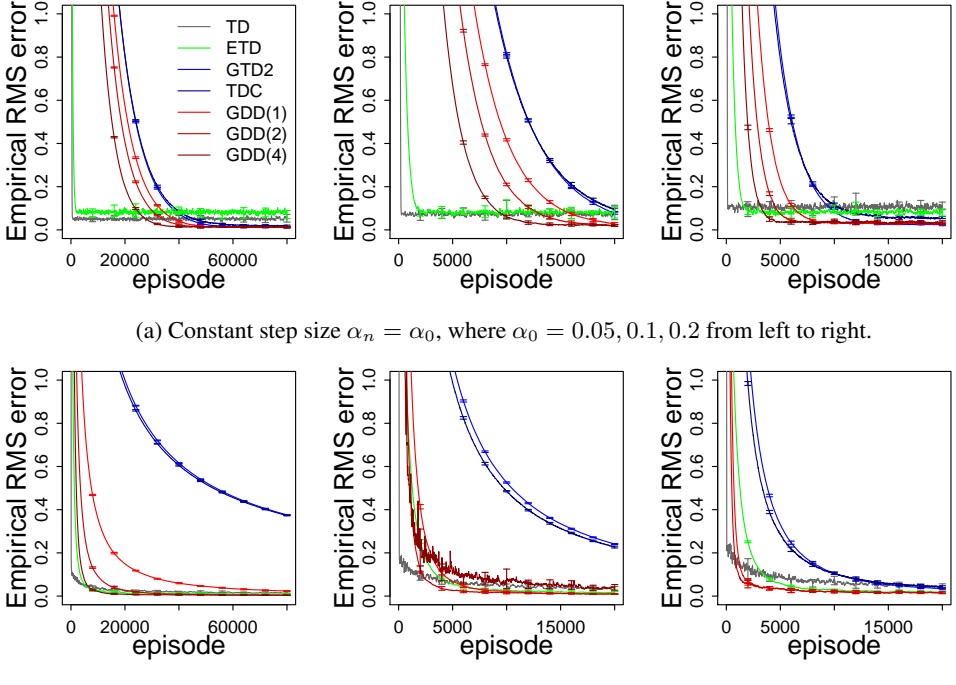

(a) Constant step size $\alpha_n = \alpha_0$, where $\alpha_0 = 0.05, 0.1, 0.2$ from left to right.

(b) Tapering step size $\alpha_n = \alpha_0(10^3 + 1)/(10^3 + n)$, where $\alpha_0 = 0.3, 0.5, 0.8$ from left to right.

Figure 4: Performance of Gradient-DD in the Boyan Chain task with 20 features. Note that the case Gradient-DD(4), i.e. $c = 4$, is not shown when it does not converge.

## 5.3 BAIRD'S COUNTEREXAMPLE

We also verify the performance of Gradient-DD on Baird's off-policy counterexample (Baird, 1995), for which TD learning famously diverges. We consider three cases: 7-state, 100-state and 500-state. We set $\alpha = 0.02$ (but $\alpha = 10^{-5}$ for ETD), $\beta = \alpha$ and $\gamma = 0.99$. We set $\kappa = 0.2$ for GDD1, $\kappa = 0.4$ for GDD2 and $\kappa = 0.8$ for GDD3. For the initial parameter values $(1, \cdots, 1, 10, 1)^\top$. We measure the performance by the empirical RMS errors as function of sweep, and report the results in Figure 5. The figure demonstrates that Gradient-DD works as well on this well-known counterexample as GTD2 does, and even works better than GTD2 for the 100-state case. We also observe that the performance improvement of Gradient-DD increases as the state spaces increases. We also note that, because the linear approximation leaves a residual error in the value estimation due to the projection

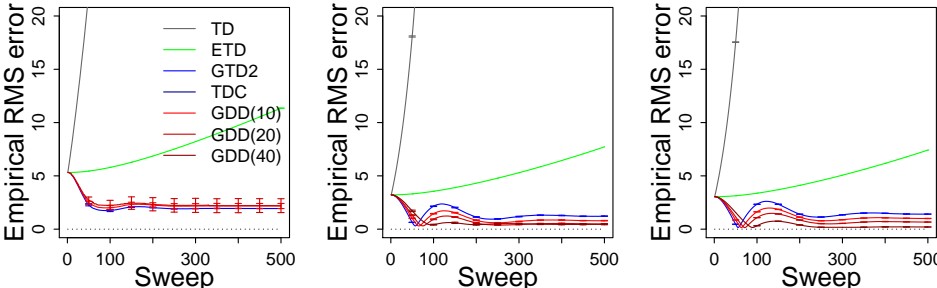

Figure 5: Bairds off-policy counterexample. From left to right: 7-state, 100-state, and 500-state. TDC is not reported here due to its similarity to GTD2. We set $\alpha = 0.02$ (but $\alpha = 10^{-5}$ for ETD), $\beta = \alpha$, and $\kappa = 0.02c$. GDD(c) denotes the Gradient-DD with $c$.

error, the RMS errors in this task do not go to zero. Interestingly, Gradient-DD reduces this residual error as the size of the state space increases.

## 6   CONCLUSION AND DISCUSSION

In this work, we have proposed Gradient-DD learning, a new gradient descent-based TD learning algorithm. The algorithm is based on a modification of the projected Bellman error objective function for value function approximation by introducing a second-order difference term. The algorithm significantly improves upon existing methods for gradient-based TD learning, obtaining better convergence performance than conventional linear TD learning.

Since GTD learning was originally proposed, the Gradient-TD family of algorithms has been extended for incorporating eligibility traces and learning optimal policies (Maei & Sutton, 2010; Geist & Scherrer, 2014), as well as for application to neural networks (Maei, 2011). Additionally, many variants of the vanilla Gradient-TD methods have been proposed, including HTD (Hackman, 2012) and Proximal Gradient-TD (Liu et al., 2016). Because Gradient-DD just modifies the objective error of GTD2 by considering an additional squared-bias term, it may be extended and combined with these other methods, potentially broadening its utility for more complicated tasks.

In this work we have focused on value function prediction in the two simple cases of tabular representations and linear approximation. An especially interesting direction for future study will be the application of Gradient-DD learning to tasks requiring more complex representations, including neural network implementations. Such approaches are especially useful in cases where state spaces are large, and indeed we have found in our results that Gradient-DD seems to confer the greatest advantage over other methods in such cases. Intuitively, we expect that this is because the difference between the optimal update direction and that chosen by gradient descent becomes greater in higher-dimensional spaces (cf. Fig. 1). This performance benefit in large state spaces suggests that Gradient-DD may be of practical use for these more challenging cases.

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

## APPENDIX

### 6.1 ON THE EQUIVALENCE OF EQNS. (7) & (8)

The Karush-Kuhn-Tucker conditions of Eqn. (8) are the following system of equations

$$\begin{cases} \frac{d}{d\mathbf{w}} J(\mathbf{w}) + \kappa \frac{d}{d\mathbf{w}}(\|\mathbf{V_w} - \mathbf{V_{w_{n-1}}}\|_{\mathbf{D}}^2 - \mu) = 0; \\ \kappa(\|\mathbf{V_w} - \mathbf{V_{w_{n-1}}}\|_{\mathbf{D}}^2 - \mu) = 0; \\ \|\mathbf{V_w} - \mathbf{V_{w_{n-1}}}\|_{\mathbf{D}}^2 \le \mu; \\ \kappa \ge 0. \end{cases}$$

These equations are equivalent to

$$\begin{cases} \frac{d}{d\mathbf{w}} J(\mathbf{w}) + \kappa \frac{d}{d\mathbf{w}} \|\mathbf{V_w} - \mathbf{V_{w_{n-1}}}\|_{\mathbf{D}}^2 = 0 \text{ and } \kappa > 0, \\ \qquad \qquad \text{if } \|\mathbf{V_w} - \mathbf{V_{w_{n-1}}}\|_{\mathbf{D}}^2 = \mu; \\ \frac{d}{d\mathbf{w}} J(\mathbf{w}) = 0 \text{ and } \kappa = 0, \text{ if } \|\mathbf{V_w} - \mathbf{V_{w_{n-1}}}\|_{\mathbf{D}}^2 < \mu. \end{cases}$$

Thus, for any $\mu > 0$, there exists a $\kappa \ge 0$ such that $\frac{d}{d\mathbf{w}} J(\mathbf{w}) + \mu \frac{d}{d\mathbf{w}} \|\mathbf{V_w} - \mathbf{V_{w_{n-1}}}\|_{\mathbf{D}}^2 = 0$.

### 6.2 EIGENVALUES OF J

Let $\lambda$ be an eigenvalue of the matrix $\mathbf{J}$. We have that

$$\begin{aligned} |\lambda\mathbf{I} - \mathbf{J}| &= \begin{vmatrix} \lambda\mathbf{I} + \sqrt{\zeta}\mathbf{G} & \kappa\mathbf{H} \\ \sqrt{\zeta}\alpha^{-1}\mathbf{G} & \lambda\mathbf{I} + \alpha^{-1}(\mathbf{I} + \kappa\mathbf{H}) \end{vmatrix} \\ &= \begin{vmatrix} \lambda\mathbf{I} + \sqrt{\zeta}\mathbf{G} & \kappa\mathbf{H} \\ -\lambda\alpha^{-1}\mathbf{I} & \lambda\mathbf{I} + \alpha^{-1}\mathbf{I} \end{vmatrix} \\ &= \begin{vmatrix} \lambda\mathbf{I} + \sqrt{\zeta}\mathbf{G} & \kappa\mathbf{H} \\ 0 & \lambda\mathbf{I} + \alpha^{-1}\mathbf{I} + \kappa\alpha^{-1}\lambda(\lambda\mathbf{I} + \sqrt{\zeta}\mathbf{G})^{-1}\mathbf{H} \end{vmatrix} \\ &= |(\lambda\mathbf{I} + \sqrt{\zeta}\mathbf{G})(\lambda\mathbf{I} + \alpha^{-1}\mathbf{I}) + \kappa\alpha^{-1}\lambda\mathbf{H}|. \end{aligned}$$

From the assumption $\mathrm{E}(\mathbf{x}_n\mathbf{x}_n^\top) = \mathbf{I}$ and the definition of $\mathbf{H}$, some eigenvalues of the matrix $\mathbf{J}$, $\lambda$, are solutions to

$$|\lambda\mathbf{I} - \mathbf{J}| = (\lambda + \lambda_G)(\lambda + \alpha^{-1}) = 0;$$

and other eigenvalues of the matrix $\mathbf{J}$, $\lambda$, are solutions to

$$\begin{aligned} |\lambda\mathbf{I} - \mathbf{J}| &= (\lambda + \lambda_G)(\lambda + \alpha^{-1}) + \kappa\alpha^{-1}\lambda \\ &= \lambda^2 + [\alpha^{-1}(1 + \kappa) + \lambda_G]\lambda + \alpha^{-1}\lambda_G = 0. \end{aligned}$$

Note $\lambda_G > 0$. the pair solutions to the equation above are

$$\begin{aligned} \lambda &= -\frac{1}{2}[\alpha^{-1}(1 + \kappa) + \lambda_G] \pm \frac{1}{2}\sqrt{[\alpha^{-1}(1 + \kappa) + \lambda_G]^2 - 4\alpha^{-1}\lambda_G} \\ &= -\frac{1}{2}[\alpha^{-1}(1 + \kappa) + \lambda_G] \pm \frac{1}{2}\sqrt{[\alpha^{-1}(1 + \kappa) - \lambda_G]^2 + 4\alpha^{-1}\lambda_G\kappa}. \end{aligned}$$

Thus, the smaller eigenvalues of the pairs are

$$\begin{aligned} \lambda_m &= -\frac{1}{2}[\alpha^{-1}(1 + \kappa) + \lambda_G] - \frac{1}{2}\sqrt{[\alpha^{-1}(1 + \kappa) - \lambda_G]^2 + 4\alpha^{-1}\lambda_G\kappa} \\ &< -\frac{1}{2}[\alpha^{-1}(1 + \kappa) + \lambda_G] - \frac{1}{2}\sqrt{[\alpha^{-1}(1 + \kappa) - \lambda_G]^2}, \end{aligned}$$

where the inequality is from $\lambda_G > 0$. When $\alpha^{-1}(1 + \kappa) - \lambda_G > 0$, then

$$\begin{aligned} \lambda_m &< -\frac{1}{2}[\alpha^{-1}(1 + \kappa) + \lambda_G] - \frac{1}{2}(\alpha^{-1}(1 + \kappa) - \lambda_G) \\ &= -\alpha^{-1}(1 + \kappa) \\ &< -\lambda_G, \end{aligned}$$

When $\alpha^{-1}(1 + \kappa) - \lambda_G \le 0$, then

$$\begin{aligned} \lambda_m &< -\frac{1}{2}[\alpha^{-1}(1 + \kappa) + \lambda_G] + \frac{1}{2}(\alpha^{-1}(1 + \kappa) - \lambda_G) \\ &= -\lambda_G, \end{aligned}$$

CONVERGENCE WITH CONSTANT STEP SIZES

At last we apply the ODE method of stochastic approximation to obtain the convergence performance.

**Theorem 1** *Consider the update rules (10) with constant step size sequences $\kappa$, $\alpha$ and $\beta$ satisfying $\kappa \geq 0$, $\beta = \zeta\alpha$, $\zeta > 0$, $\alpha \in (0,1)$ and $\beta > 0$. Let the TD fixed point be $\mathbf{w}^*$, such that $\mathbf{V}_{\mathbf{w}^*} = \mathbf{\Pi}\mathbf{B}\mathbf{V}_{\mathbf{w}^*}$. Suppose that (A1) $(\mathbf{x}_n, r_n, \mathbf{x}_{n+1})$ is an i.i.d. sequence with uniformly bounded second moments, and (A2) $E[(\mathbf{x}_n - \gamma\mathbf{x}_{n+1})\mathbf{x}_n^\top]$ and $E(\mathbf{x}_n\mathbf{x}_n^\top)$ are non-singular. Then for any $\epsilon > 0$, there exists $b_1 < \infty$ such that*

$$\limsup_{n\to\infty} P(\|\mathbf{w}_n - \mathbf{w}^*\| > \epsilon) \leq b_1\alpha.$$

**Proof** From the constant step sizes in the conditions, we denote $\kappa_n = \kappa$ and $\alpha_n = \alpha$. Thus, Eqn. (12) equals

$$(\mathbf{I} + \kappa\mathbf{H}_n)(\boldsymbol{\rho}_{n+1} - \boldsymbol{\rho}_n) - \kappa\mathbf{H}_n(\boldsymbol{\rho}_{n+1} - 2\boldsymbol{\rho}_n + \boldsymbol{\rho}_{n-1})$$
$$= -\sqrt{\zeta}\alpha(\mathbf{G}_n\boldsymbol{\rho}_n - \mathbf{g}_{n+1}). \tag{A.1}$$

Denoting $\boldsymbol{\psi}_{n+1} = \alpha^{-1}(\boldsymbol{\rho}_{n+1} - \boldsymbol{\rho}_n)$, Eqn. (A.1) is rewritten as

$$\begin{bmatrix} \boldsymbol{\rho}_{n+1} - \boldsymbol{\rho}_n \\ \boldsymbol{\psi}_{n+1} - \boldsymbol{\psi}_n \end{bmatrix}$$
$$= \alpha \begin{bmatrix} \mathbf{I} + \kappa\mathbf{H}_n & -\kappa\alpha\mathbf{H}_n \\ \mathbf{I} & -\alpha\mathbf{I} \end{bmatrix}^{-1} \begin{bmatrix} -\sqrt{\zeta}(\mathbf{G}_n\boldsymbol{\rho}_n - \mathbf{g}_{n+1}) \\ \boldsymbol{\psi}_n \end{bmatrix}$$
$$= \alpha \begin{bmatrix} -\sqrt{\zeta}\mathbf{G}_n & -\kappa\mathbf{H}_n \\ -\sqrt{\zeta}\alpha^{-1}\mathbf{G}_n & -\alpha^{-1}(\mathbf{I} + \kappa\mathbf{H}_n) \end{bmatrix} \begin{bmatrix} \boldsymbol{\rho}_n \\ \boldsymbol{\psi}_n \end{bmatrix}$$
$$+ \alpha \begin{bmatrix} \sqrt{\zeta}\mathbf{g}_{n+1} \\ \sqrt{\zeta}\alpha^{-1}\mathbf{g}_{n+1} \end{bmatrix}, \tag{A.2}$$

where the second step is from

$$\begin{bmatrix} \mathbf{I} + \kappa\mathbf{H}_n & -\kappa\alpha\mathbf{H}_n \\ \mathbf{I} & -\alpha\mathbf{I} \end{bmatrix}^{-1} = \begin{bmatrix} \mathbf{I} & -\kappa\mathbf{H}_n \\ \alpha^{-1}\mathbf{I} & -\alpha^{-1}(\mathbf{I} + \kappa\mathbf{H}_n) \end{bmatrix}.$$

Denoting $\mathbf{G} = \mathrm{E}(\mathbf{G}_n)$, $\mathbf{g} = \mathrm{E}(\mathbf{g}_n)$ and $\mathbf{H} = \mathrm{E}(\mathbf{H}_n)$, then the TD fixed point of Eqn. (A.1) is given by

$$-\mathbf{G}\boldsymbol{\rho} + \mathbf{g} = 0 \tag{A.3}$$

We apply the ordinary differential equation approach of the stochastic approximation in Theorem 1 (Theorem 2.3 of (Borkar & Meyn, 2000)) into Eqn. (A.2). Note that (Sutton et al., 2009a) and (Sutton et al., 2009b) also applied Theorem 2.3 of (Borkar & Meyn, 2000) in using the gradient-descent method for temporal-difference learning to obtain their convergence results. For simplifying notation, denote

$$\mathbf{J}_n = \begin{bmatrix} -\sqrt{\zeta}\mathbf{G}_n & -\kappa\mathbf{H}_n \\ -\sqrt{\zeta}\alpha^{-1}\mathbf{G}_n & -\alpha^{-1}(\mathbf{I} + \kappa\mathbf{H}_n) \end{bmatrix},$$

$$\mathbf{J} = \begin{bmatrix} -\sqrt{\zeta}\mathbf{G} & -\kappa\mathbf{H} \\ -\sqrt{\zeta}\alpha^{-1}\mathbf{G} & -\alpha^{-1}(\mathbf{I} + \kappa\mathbf{H}) \end{bmatrix},$$

$\mathbf{y}_n = \begin{bmatrix} \boldsymbol{\rho}_n \\ \boldsymbol{\psi}_n \end{bmatrix}$, $\mathbf{h}_n = \begin{bmatrix} \sqrt{\zeta}\mathbf{g}_{n+1} \\ \sqrt{\zeta}\alpha^{-1}\mathbf{g}_{n+1} \end{bmatrix}$, and $\mathbf{h} = \begin{bmatrix} \sqrt{\zeta}\mathbf{g} \\ \sqrt{\zeta}\alpha^{-1}\mathbf{g} \end{bmatrix}$. Eqn. (A.2) is rewritten as

$$\mathbf{y}_{n+1} = \mathbf{y}_n + \alpha(f(\mathbf{y}_n) + \mathbf{h} + \mathbf{M}_{n+1}), \tag{A.4}$$

where $f(\mathbf{y}_n) = \mathbf{J}\mathbf{y}_n$ and $\mathbf{M}_{n+1} = (\mathbf{J}_n - \mathbf{J})\mathbf{y}_n + \mathbf{h}_n - \mathbf{h}$.

Now we verify the conditions (c1-c4) of Lemma 1. Firstly, Condition (c1) is satisfied under the assumption of constant step sizes. Secondly, $f(\mathbf{y})$ is Lipschitz and $f_\infty(\mathbf{y}) = \mathbf{G}\mathbf{y}$. Following Sutton et al. (2009a), the Assumption A2 implies the real parts of all the eigenvalues of $\mathbf{G}$ are positive. Therefore, Condition (c2) is satisfied.

Because $E(\mathbf{M}_{n+1}|\mathcal{F}_n) = 0$ and $\mathrm{E}(\|\mathbf{M}_{n+1}\|^2|\mathcal{F}_n) \leq c_0(1+\|\mathbf{y}_n\|^2)$, where $\mathcal{F}_n = \sigma(\mathbf{y}_i, \mathbf{M}_i, i \leq n)$, is a martingale difference sequence, we have that

$$\|\mathbf{M}_{n+1}\|^2 \leq 2(\|\mathbf{J}_n - \mathbf{J}\|^2\|\mathbf{y}_n\|^2 + \|\mathbf{h}_n - \mathbf{h}\|^2). \tag{A.5}$$

From the assumption A1, Eqn. (A.5) follows that there are constants $c_j$ and $c_h$ such that

$$\mathrm{E}(\|\mathbf{J}_n - \mathbf{J}\|^2|\mathcal{F}_n) \leq c_j;$$
$$\mathrm{E}(\|\mathbf{h}_{n+1} - \mathbf{h}\|^2) \leq c_h.$$

Thus, Condition (c3) is satisfied.

Finally, Condition (c4) is satisfied by noting that $\mathbf{y}^* = \mathbf{G}^{-1}\mathbf{g}$ is the unique globally asymptotically stable equilibrium. ∎

Theorem 1 bounds the estimation error of $\mathbf{w}$ in probability. Note that the convergence of Gradient-DD learning provided in Theorem 1 is a somewhat weaker result than the statement that $\mathbf{w}_n \to \mathbf{w}^*$ with probability 1 as $n \to \infty$. The technical reason for this is the condition on step sizes. In Theorem 1, we consider the case of constant step sizes, with $\alpha_n = \alpha$ and $\kappa_n = \kappa$. This restriction is imposed so that Eqn. (12) can be written as a system of first-order difference equations, which cannot be done rigorously when step sizes are tapered as in (Sutton et al., 2009b). As shown below, however, we find empirically in numerical experiments that the algorithm does in fact converge with tapered step sizes and even obtains much better performance in this case than with fixed step sizes.

AN ODE RESULT ON STOCHASTIC APPROXIMATION

We introduce an ODE result on stochastic approximation in the following lemma, then prove Theorem 1 by applying this result.

**Lemma 1** *(Theorem 2.3 of Borkar & Meyn (2000)) Consider the stochastic approximation algorithm described by the $d$-dimensional recursion*

$$\mathbf{y}_{n+1} = \mathbf{y}_n + a_n[f(\mathbf{y}_n) + \mathbf{M}_{n+1}].$$

*Suppose the following conditions hold: (c1) The sequence $\{\alpha_n\}$ satisfies for some constant $0 < \underline{\alpha} < \bar{\alpha} < 1$, $\underline{\alpha} < \alpha_n < \bar{\alpha}$; (c2) The function $f$ is Lipschitz, and there exists a function $f_\infty$ such that $\lim_{r \to \infty} f_r(\mathbf{y}) = f_\infty(\mathbf{y})$, where the scaled function $f_r : \mathbb{R}^d \to \mathbb{R}^d$ is given by $f_r(\mathbf{y}) = f(r\mathbf{y})/r$. Furthermore, the ODE $\dot{\mathbf{y}} = f_\infty(\mathbf{y})$ has the origin as a globally asymptotically stable equilibrium; (c3) The sequence $\{\mathbf{M}_n, \mathcal{F}_n\}$, with $\mathcal{F}_n = \sigma(\mathbf{y}_i, \mathbf{M}_i, i \leq n)$, is a martingale difference sequence. Moreover, for some $c_0 < \infty$ and any initial condition $y_0$, $E(\|\mathbf{M}_{n+1}\|^2|\mathcal{F}_n) \leq c_0(1 + \|\mathbf{y}_n\|^2)$. (c4) The ODE*

$$\dot{\mathbf{y}}(t) = f(\mathbf{y}(t))$$

*has a unique globally asymptotically stable equilibrium $\mathbf{y}^*$. Then for any $\epsilon > 0$, there exists $b_1 < \infty$ such that $\limsup_{n \to \infty} P(\|\mathbf{y}_n - \mathbf{y}^*\| > \epsilon) \leq b_1\bar{\alpha}$.*

6.3 ADDITIONAL EMPIRICAL RESULTS

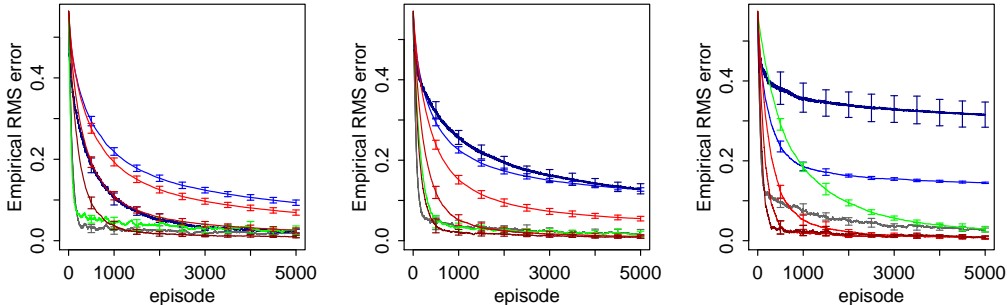

Figure 6: Performance of Gradient-DD in the random walk task under Case 3. Tapering step size $\alpha_n = \alpha_0(10^3 + 1)/(10^3 + n)$ and $\kappa$ is allowed to be dependent on $n$: $\kappa_n = c\alpha_n$. From left to right in each subfigure: the size of state space is 10 ($\alpha_0 = 0.1$), 20 ($\alpha_0 = 0.2$), 50 ($\alpha_0 = 0.5$). Results are averaged over 20 runs, with error bars denoting standard deviations across runs.

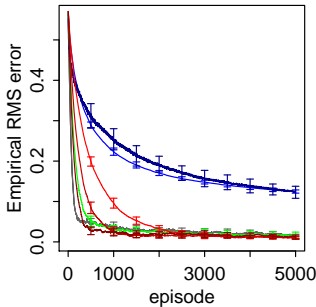

Figure 7: Performance of Gradient-DD in the random walk task when the initial values are set 0. The size of state space is 20, with tapering step size $\alpha_n = 0.2(10^3 + 1)/(10^3 + n)$. Results are averaged over 20 runs, with error bars denoting standard deviations across runs.

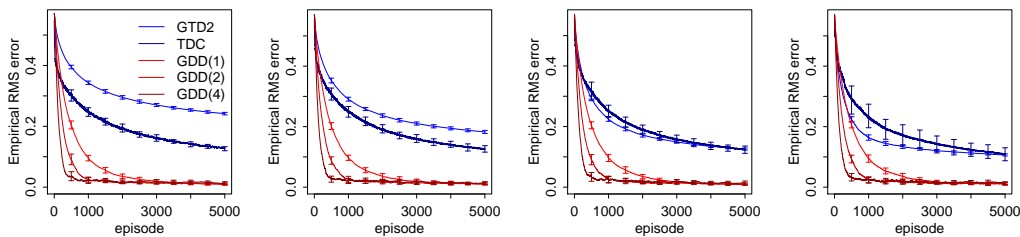

Figure 8: Performance under various $\beta_n$ in the random walk task with 20 states. $\alpha_n = \alpha_0(10^3 + 1)/(10^3 + n)$ with $\alpha_0 = 0.2$. From left to right in each subfigure: the size of state space is $\beta_n = \alpha_n/4$, $\beta_n = \alpha_n/2$, $\beta_n = \alpha_n$, and $\beta_n = 2\alpha_n$. Results are averaged over 20 runs, with error bars denoting standard deviations across runs.

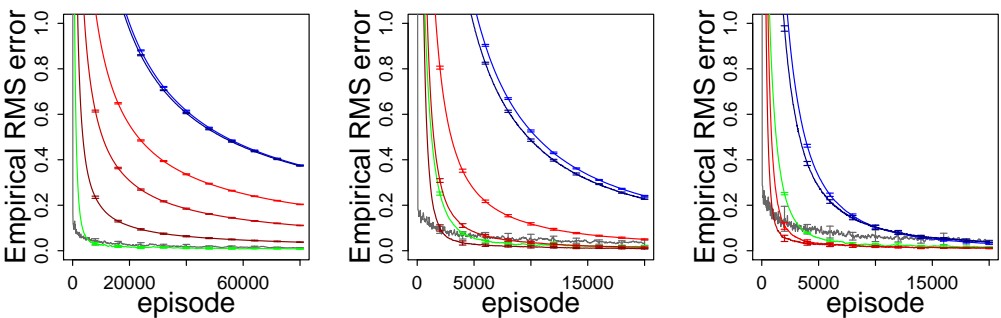

Figure 9: Performance of Gradient-DD in the Boyan Chain task with 20-features under Case 3. Tapering step size $\alpha_n = \alpha_0(10^3+1)/(10^3+n)$, where $\alpha_0 = 0.3, 0.5, 0.8$, , where $\alpha_0 = 0.3, 0.5, 0.8$ from left to right, and $\kappa$ is allowed to be dependent on $n$: $\kappa_n = c\alpha_n$ Note that the case GDD(4), i.e. $c = 4$, is not shown when it does not converge.

