# OpenReview forum: "Gradient descent temporal difference-difference learning"
_ICLR.cc/2021/Conference — Reject_

### Official Review · AnonReviewer1 · 2020-10-15
**Official Blind Review #1**

**Rating:** 3
**Confidence:** 4

**Review:**

## Summary of paper
This paper introduces a novel regularized mean-squared projected bellman objective and corresponding GTD2-like algorithm which minimizes the objective. The paper analytically investigates the convergence rate of the proposed algorithm, then empirically investigates the performance of the algorithm across several problems with linear function approximation.

## Summary of review
This paper is a clear reject for me. There appear to be significant issues in both the analytical section and the empirical section; which in total bring into question the utility of the proposed algorithm. The literature review also appears to be lacking and there appear to be several minor incorrect statements throughout the paper. I feel quite confident in my evaluation of the empirical section and literature review. I feel confident that there is a bug/typo/incorrect result in the analytical section. I did not attempt to debug the proof to determine which of the three (bug, typo, or incorrect result) was true. Details follow roughly in order of greatest concern -> least concern.

### Proof of convergence rate
I follow the proof up to equation (12). This is a standard result from (e.g) Maei's 2011 thesis. After equation (12) I follow the transformation that solved for $\rho_{n+1} - \rho_n$ resulting in a matrix inverse. However, after distributing the matrix inverse, I remain confused why $\sqrt{\xi} g_{n+1}$ is not multiplied by the inverted matrix? This, however, may not change the resulting eigenvalues so I do not believe it will change the result. Continuing on to equation (14), I did not check for correctness solving the polynomial for the eigenvalues, so I will rely on the given result in the paper. The conclusion of which states: $\lambda = - \frac{1}{2} (\alpha^{-1} + \lambda_G) \pm \frac{1}{2} \sqrt{\text{thing}} < -\lambda_G$. Ignoring a few details in the middle, the result ultimately states $\lambda < -\lambda_G$. Here lies the fundamental problem. Because GTD2 is known to be a convergent linear system, then we know $\lambda_G$ are all strictly negative (proof of this in Maei's thesis) under some assumptions (notably the invertibility of $X X^\top$). For such a linear system to be convergent, we **must** have the real part of the eigenvalues be strictly negative. The proof under equation (14) states $\lambda < -\lambda_G$, which because $\lambda_G$ is strictly negative, means that $\lambda < \text{some positive number}$. This (a) tells us nothing of the comparative convergence rates since $\lambda$ could be larger than $\lambda_G$, and (b) also suggests that there could be cases where the proposed algorithm does not even converge in the first place because $\lambda$ could be greater than or equal to 0. This could be a typo and the negative sign on the right shouldn't exist, but because there are few details from equation (14) to the end, I was unable to debug and decide if this is typo/bug/incorrect conclusion.

There a few other issues with the proof that concern me.
* Do we know that the upper-bound on $\kappa$ is reasonable? The quantity $\alpha (\lambda_G - \alpha^{-1})^2 / 4$ is difficult to interpret, but the reliance on $\alpha^{-1}$ seems to imply a preference toward much smaller stepsizes; which seems counter-intuitive towards the goal of improving convergence rates.
* Likewise, the first part of the solution for $\lambda$ we have $-\frac{1}{2} (\alpha^{-1} - \lambda_G)$ which likewise implies that smaller stepsizes significantly improve convergence rate (small $\alpha$ implies highly negative $\lambda$). Considering the remainder of the solution for $\lambda$ is under a square-root, this first factor appears to be dominant (though I could be wrong on this, more insight would be appreciated!). A stepsize approaching 0 would yield the fastest convergence rate (apparently) by having the smallest eigenvalue approach negative infinity. This simply does not pass the "smell test".
* Further building on my previous point, for large enough values of $\alpha$ we will be in a situation where $\alpha^{-1} < |\lambda_G|$ meaning that the first term will become positive. Considering that the second term is a plus-or-minus, then we could be adding a positive term to a positive term yielding a positive eigenvalue. This means that for large enough $\alpha$ we don't have convergence any longer. I wonder how feasible the upper-bound on $\alpha$ is to guarantee convergence. I would also like to see these assumptions explicitly stated in the proof.
* A lot of little details were left out of the proof. Where are the assumptions on boundedness of the features and the rewards? Can one show that the noise sequence of the modified algorithm is actually a Martingale Difference Sequence and thus the result from Borkar and Meyn 2000 holds? Need there be an assumption of independent samples or are these samples coming from Markovian sampling?

### Empirical section
* The choice to set the initial value function $V(S) = 0.5 \forall S$ for the random walk was odd. I suppose that because the left reward = 0 and the right reward = 1, and the policy is 50% chance to go left or right, and $\gamma=1$, then the optimal value function $v_\pi$ linearly interpolates from (0, 1) with the center-most state having value $V(n // 2) = 0.5$. This choice seems likely to disproportionately favor the proposed GDD algorithm; which encourages the value function estimate to change slowly. Because the initial estimates are so close to correct, only small changes will be necessary and the regularizer term will remain small. What happens if the value function is initialized to 0 everywhere, or even to -1 everywhere?
* The exclusion of TDC from the stepsize sensitivity investigation makes little sense to me. The first experiment chose an aggressively large stepsize $\alpha = 0.5$ for which TDC performed poorly. Then did not investigate the sensitivity of TDC to stepsize in later plots because of this choice. If you check [Giassian et al. 2020], they report that TDC in fact out-performs GTD2 on all of the same domains tested here for appropriately chosen $\alpha$ and $\beta$.
* The choice of $\beta$ is never discussed. How did you set $\beta$?
* How many runs? What is the variance? Are any results statistically significant?
* The primary motivation of the paper was around off-policy learning, yet only one of the tested domains was off-policy (Baird's Counterexample star MDP). It would have been nice to see the random walks made into off-policy domains.

### Literature review
* This paper modifies the MSPBE by adding a regularizer term. There are a few other papers in the literature that do this and derive the corresponding GTD2/TDC algorithms. Liu et al. 2012 and Ghiassian et al. 2020 immediately come to mind. These should both be cited and discussed.
* Are there any papers that add such a constraint as $\| w_n - w_{n - 1} \|^2$ to any known objective function? This seems like an odd choice of regularizer (penalizes making changes to the weights), so any prior literature from any field (supervised learning, online learning, optimization, etc.) would go a long way in convincing the reader that this is a good idea.
* The paper mentions several times that GTD methods converge more slowly than TD. I know of a single proof that shows this in Maei's thesis for the GTD algorithm. I do not know of any such proof for TDC or GTD2. There exists empirical evidence of this in Ghiassian et al. 2020 or White and White 2016, but neither of these papers are cited.
* GTD methods and importance sampling are not mutually exclusive methods for off-policy learning. In fact, GTD methods canonically use IS for their off-policy variants. Further importance sampling definitely does not decrease the variance of parameter updates (mentioned in the second paragraph of Section 1).
* Sutton et al. 2009 is not really a breakthrough in the study of convergence properties of MDP systems. In fact, the proofs of Sutton et al. 2009 do not even assume samples are drawn from a distribution induced by an MDP. Perhaps Borkar and Meyn 2000 is a better reference as it fundamentally builds the proof structure used by Sutton et al. 2009?

### Other minutiae
* Eye-balling the modified objective function leads me to believe the objective shares the same fixed-point as the MSPBE (thus the new GTD2 algorithm converges to the same fixed-point as TD), but it would be nice to show this formally in the analytical section.
* Is it possible to extend this objective to the non-linear setting?
* The paper mentions that the proposed regularizer avoids large biases in the updating process. Does it not *add* bias to the updating process? Perhaps it was meant that the regularizer avoids high variance? Either way, a careful analytical discussion of the bias-variance properties would go a long way towards improving this paper.
* In section 5.3 what is $\eta$? I believe this is supposed to be $\beta$ (i.e. the stepsize for the secondary weights) since $\eta$ is your secondary weight vector?
* Would it make more sense to consider $\kappa$ to be a regularizer parameter instead of a stepsize and having it absorb $\alpha$? It seems in the experiment section you split these anyways, so perhaps it makes the analytical section much more clear if the algorithm was instead $\alpha\kappa (x^\top w_n - x^\top w_{n-1})$.
* The paper repeatedly defines off-policy learning as learning the optimal policy using an exploratory policy. This is a bit of a restrictive setting and is certainly not the setting that Sutton et al. 2009 considered (the work that this paper builds upon).
* Why assume that the target policy is deterministic (mentioned in Section 2.1)? This is a strange choice that is not used in either the empirical or analytical section as far as I can tell.
* It is mentioned that TD methods should seek to minimize the MSPBE (or perhaps the MSBE it isn't clear which is meant), but shouldn't instead the goal be to minimize the MSVE (e.g. $\| \hat{V}_w - v_\pi \|^2$)?

### Papers mentioned in this review
Maei, Hamid Reza. “Gradient Temporal-Difference Learning Algorithms.” University of Alberta, 2011.

Sina Ghiassian, Andrew Patterson, Shivam Garg, Dhawal Gupta, Adam White, and Martha White. “Gradient Temporal-Difference Learning with Regularized Corrections.” International Conference on Machine Learning, 2020. http://arxiv.org/abs/2007.00611.

Adam White, and Martha White. “Investigating Practical Linear Temporal Difference Learning.” International Conference on Autonomous Agents and Multi-Agent Systems, 2016. http://arxiv.org/abs/1602.08771.

Borkar, V. S., & Meyn, S. P. (2000). The O.D.E. Method for Convergence of Stochastic Approximation and Reinforcement Learning. SIAM Journal on Control and Optimization, 38(2), 447–469. https://doi.org/10.1137/S0363012997331639

Sutton, R. S., Maei, H. R., Precup, D., Bhatnagar, S., Silver, D., Szepesvári, C., & Wiewiora, E. (2009). Fast gradient-descent methods for temporal-difference learning with linear function approximation. Proceedings of the 26th Annual International Conference on Machine Learning - ICML ’09, 1–8. https://doi.org/10.1145/1553374.1553501

Liu, B., Mahadevan, S., & Liu, J. (2012). Regularized Off-Policy TD-Learning. Advances in Neural Information Processing Systems, 9.

# After discussion and edits

I acknowledge that I have read the other reviews and resulting discussions and I have read the relevant changes in the edited text. I have raised my score from 2->3 to reflect that several concerns were alleviated through the edits, but several new concerns (and old concerns) remain. I will summarize below.

After the author edits, the issue with convergence and convergence rates appear to have been resolved. I additionally appreciate the much greater clarity in the analytical section. However, I still find the contribution to be borderline at best in terms of novelty of approach and I find that the evidence of applicability is still considerably lacking. The introduction of a regularizer to accelerate GTD methods is itself not novel. The form of the proposed regularizer is novel, however, I find its form to be unintuitive as it punishes making changes to the weights. There are some prior works that motivate this well (i.e. TRPO and other trust-region optimization techniques), but this paper does not appeal to prior works to motivate their regularizer. Instead, I must rely on the empirical study which does not investigate the learning speed of the proposed algorithm compared to baselines. In many cases, the proposed algorithm does not clearly outperform baselines.

---

> ### Author Response · Authors · 2020-11-24
> **clarify the convergence analysis and empirical studied are updated**
>
> < Proof of convergence rate>
>
> - Re: This quantity is in fact multiplied by the inverted matrix. Before multiplying by the inverted matrix, it’s $(\sqrt{ξ}g_{n+1}^{\top}, 0)^{\top}$. After the multiplying by the inverted matrix, it becomes
> $(\sqrt{ξ}g_{n+1}^{\top}, \sqrt{ξ}\alpha^{-1}g_{n+1}^{\top})^{\top}$.
> The G_n in this paper is the negative G_n of the Maei’s thesis, since we directly use the gradient but do not change the sign. Thus, the eigenvalues lambda from our G are strictly positive. We have clarified this point and added additional details about deriving the eigenvalues in the main text.
>
> <other issues with the proof>
>
> - Re:  We can write $\alpha(\lambda_G−\alpha^{-1})^2/4=(\alpha^{-1/2}−\lambda_G\alpha^{1/2})^2/4$. When alpha goes to 0 (as it always does when n increases under the assumption of tapering step size), this upper-bound becomes large as n increases.  We have included a note to emphasize this in the main text.
>
> - Re: Because α appears in both of the terms that make up λ, it is not reasonable to draw conclusions about λ by considering each term separately. In fact, combing two parts is the solution of λ. And this is to show λ is more negative -λ_G (note λ_G is positive here), as we show in detail in the appendix.
>
> - Re: This appears to be related to an earlier point of confusion about the sign of λ_G. Because this quantity is positive in our paper, the reviewer need not be concerned about the sign of the eigenvalue changing.
>
> - Re: In response to the reviewer’s comment, we have provided a more detailed derivation in the appendix. Thanks also for the suggestion of the reference, which we have included in the updated version.
>
> < Empirical section>
>
> - Re: In response to the reviewer’s question, we have run these simulations again with the value estimate initialized to 0. The performance is similar to the case shown in the main text.
>
> - Re: Thanks for this suggestion. We have done the stepsize sensitivity on TDC, and have chosen the appropriate step sizes for TDC.
>
> - Re: In the version that we originally submitted, we set \beta=\alpha. We apologize for neglecting to mention this explicitly. In the updated version, we have made this explicit, and we have also added a new figure showing the effect of beta not equal to alpha in the Random Walk task.
>
> - Re: Thanks for this suggestion. In the revised version, we report the number of runs and standard deviation error bars for all of our empirical results.
>
> - Re: We thank the reviewer for the suggestion of the random walks made into off-policy domains, and aim to do this in a future extension work.
>
> < Literature review>
>
> - Re: Thanks to the reviewer for these references. We have included citations to them in the revised version, and re-written the related paragraphs.
>
> <Other minutiae>
>
> - for the comment of show that the objective shares the same fixed-point as the MSPBE,
> Re: We have implemented this suggestion in the revised version of our manuscript.
>
> - for the comment of "extend this objective to the non-linear setting",
> Re: Yes, it is possible to follow the GTD methods in non-linear setting. We are working on this and plan to present the results in a follow-up paper.
>
> - for the comment on the word "biases",
> Re: Thanks to the reviewer for this comment. We agree that the term “bias” in this context is misleading and have used the term “drastic changes” in the place of “biases” in the updated version.
>
> - Re: \eta in Section 5.3 is a typo. Sorry for that. It should have been zeta (the ratio of beta and alpha). We have modified it.
>
> - for the comment of notations \alpha and \kappa,
> Re: While we appreciate the suggestion and recognize that there are tradeoffs associated with different notational choices, we have decided to keep the notation from our earlier version. In our view at least, this makes the analytical section, in which step sizes are constant, clearer.
>
> - for the comment of statement on off-policy,
> Re: Thanks much for this point. We have re-written it.
>
> - for the assumption that target policy is deterministic,
> Re: Thanks to the reviewer for pointing this out. In fact, we never made use of this assumption, so we have dropped it in the updated version.

---

> > ### Comment · AnonReviewer1 · 2020-11-25
> > **Feedback**
> >
> > I appreciate the significant effort put towards the analytical section and cleaning up these results. I am happy that the convergence rate results appears to hold and that the edits in the paper makes this far more clear to me. The added citations for the theoretical component make it more clear, and though I do still prefer that assumptions are stated directly in the paper, the citation to Borkar and Meyn, as well as Maei, help the interested reader discover the underlying assumptions for which this proof holds. Considering this, I would raise my recommendation to a borderline score (i.e. a 6) based solely on the theoretical contribution.
> >
> > However, there are still concerns for the empirical setting; many of which surfacing with the added empirical results. In the stepsize sensitivity plots (Figure 2), it is unclear that there is a consistent winner among algorithms; other than perhaps TD. Considering that GDD gets 3 chances to win compared to other algorithms, this raises some question of utility of the proposed algorithm. Notice also that ETD is actually much less sensitive than almost any other algorithm, not more sensitive. Its sensitivity curve is shifted, but has a wider "bowl" implying that there is a larger range of admissable stepsizes.
> >
> > The learning curves in Figure 3 don't seem to match the conclusions in Figure 2. The caption of Figure 3 suggests that sub-optimal stepsizes were chosen for the competitor algorithms, while the optimal stepsize for the proposed algorithm is used. Perhaps a more fair comparison in the learning curves would be to let each algorithm pick its best stepsize (or second best perhaps to reduce maximization bias).
> >
> > Considering that Ghiassian et al. 2020 also introduce a regularization technique for gradient td methods with the explicit purpose of accelerating TDC, I would argue that it is necessary to compare to their proposed algorithm. The algorithm proposed here seems to roughly match GTD2 in performance, but is generally outperformed by TD. Contrarily, TDRC (the algorithm from Ghiassian et al.) performs roughly equivalently to TD and significantly outperforms GTD2 and TDC while still being convergent. This strongly implies to me that TDRC would significantly outperform the proposed algorithm. However, the proposed regularization technique can very likely be combined with TDRC; possibly resulting in an even superior performing algorithm.
> >
> > The performance on Baird's counterexample seems suspect to me. Both GTD2 and TDC are known to be able to converge to 0 error. Note that the comment that 0 error is unachievable due to projection error is incorrect (either that, or a different form of Baird's counterexample is being used compared to what is generally used in the literature). An optimal weight vector for Baird's is the zero vector; which is contained within the linear function class for all algorithms.
> >
> > Based on the clarifications to the theoretical section, I intend to raise my scoring of the paper. However, based on lingering empirical concerns and that the proposed algorithm does not appear to outperform its competitors even on simplistic on-policy, tabular random walk domains, Boyan's Chain, or Baird's Counterexample, I do still intend to recommend reject for this paper in its current state.

---

> > > ### Author Response · Authors · 2020-11-25
> > > **correct a mistake of derivation of $\lambda$ and replies to the empirical studies**
> > >
> > > We are grateful to the reviewer for this encouraging feedback and improving the score.
> > > We just corrected a mistake.
> > > In the derivation of the eigenvalues of J, we missed a $\lambda$ in the equation, so the correct one is as follows:
> > > $|\lambda \mathbf{I}-\mathbf{J}|=|(\lambda \mathbf{I}+ \sqrt{\zeta}\mathbf{G})(\lambda \mathbf{I}+\alpha^{-1}\mathbf{I})
> > > +\kappa\alpha^{-1}\lambda\mathbf{H}|$
> > > (Note the older and wrong one in the Appendix was: $|\lambda \mathbf{I}-\mathbf{J}|=|(\lambda \mathbf{I}+ \sqrt{\zeta}\mathbf{G})(\lambda \mathbf{I}+\alpha^{-1}\mathbf{I})
> > > +\kappa\alpha^{-1}\mathbf{H}|$).
> > > But it has little trouble on the comparison of eigenvalues of G and J. The results still hold, and the condition on $\kappa$ is not necessary any longer.  We have updated the revised manuscript. Please check it.
> > >
> > > - In Figure 2, we use the final performance as the measure for showing the performance  as a function of alpha. As we know, these algorithms depend on step sizes. If the stepsizes are too big, these algorithms would be diverge; while if step sizes are too small, the learning is too slow. Thus, our aim of plotting this figure is to show the overall performance.  From the Figure 2, we can observe the the domains of alpha for GDD and GTD2 almost overlap.
> > >
> > > - For the Baird's counterexample, we measure the error by the mean squared error rather than PBE.  As shown Figure 11.5 in Section 11.7 of Sutton \& Barto 2018,  they converges to zero PBE, but do not converge to the mean squared error (denoted VE in that book). So our results are consistent to theirs.
> > >
> > > - We will further answer other comments, specially the parameter settings,  in the subsequent revision.
> > > Thanks much again!

---

### Official Review · AnonReviewer4 · 2020-10-28
**Nice idea! The analysis is not rigorous and convincing.**

**Rating:** 5
**Confidence:** 4

**Review:**

This paper proposes a variant of the GTD2 algorithm by adding an additional regularization term to the objective function, and the new algorithm is named as Gradient-DD (GDD). The regularization ensures that the value function does not change drastically between consecutive iterations. The authors show that the update rule of GDD can be written as a difference equation and aim to further show the convergence via Lyapunov based analysis. An simulation study is provided to compare the proposed GDD algorithm with TD, ETD, and GTD.

The paper is well written in general. The idea of extra regularization on the distance between two value functions sounds reasonable to me since it resembles the constraint in trust region optimization for policy gradient methods. However, the claimed improved convergence over GTD is not rigorously proved and thus not convincing.

In Section 4, the convergence analysis is not derived in a rigorous way. It would help the readers to understand the improved convergence if the authors could complete the analysis and show the convergence rate.

Why can the eigenvalues of matrix J_n be written as the block matrix before eq (14)? It seems to me that G and H are diagonal matrices with the diagonal elements being the eigenvalues of G_n and H_n. Ideally the eigenvalues of J_n, which is denoted as J in this paper, should also be a diagonal matrix. Furthermore, since G_n is not symmetric, G may have some complex values as its eigenvalues. This is ignored from the current analysis without any explanation.

In the experiment part, Figure 3 shows that the RMS error of the GDD algorithm will blow up when step size is large. It seems that the proposed algorithm may not be as robust as the conventional TD algorithm?

#########Edits after the rebuttal#########
Thank you for the responses. After reading them and the discussion with other reviewers, I still think the current contribution of this paper is marginal and I keep my score as 5.

---

> ### Author Response · Authors · 2020-11-24
> **convergence analysis is updated**
>
> Re: We completely agree with the reviewer about this important point. Indeed, before submitting this manuscript, we invested a great deal of effort in attempting to generalize the convergence result of GTD2 to our algorithm, following the ODE approach of Borkar and Meyn (2000). This approach turns out to lead to technical difficulties. Due to the fact that our algorithm is a second-order difference equation, we are not able to implement the assumption of a tapering step size, which is essential to obtaining the strong convergence results of Borkar and Meyn and GTD2. The result is that we have been able to obtain only a somewhat weaker convergence result. In response to the comments from this reviewer and other reviewers, we have decided to include the proof of this weak convergence result, which we had omitted from the original submitted version, as an appendix in the revised version of the manuscript. While this result is not entirely satisfying, we would also point out that, empirically, Gradient-DD works well or even better in the case of tapering step size compared with the case of constant step size, suggesting that a stronger convergence proof may be possible in future work.
>
>
> We apologize for the confusion. We neglected to mention that we consider the simple case where the matrix E(x_n x_n^{\top})=I. So the eigenvalues of matrix J_n can be written as some function of \lambda_G.  We have added details in the appendix.
> Regarding the potentially complex eigenvalues of G, Sutton et al (2009) have proven that G is non-singular and the eigenvalues of G are positive. (Note that our G is defined with opposite sign relative to G from that paper.)
>
> We agree with the reviewer’s observation that the window of alpha_0 values leading to good performance may be slightly smaller for our algorithm relative to conventional TD learning for this task. Given that the window sizes are at least in the same ballpark, we hope the reviewer will agree that this is a reasonable price to pay for obtaining better convergence (Fig. 3) and stability (as shown in Baird’s counterexample).

---

### Official Review · AnonReviewer2 · 2020-10-28

**Rating:** 5
**Confidence:** 4

**Review:**

### Summary of Contributions

The paper proposes the gradient descent TD difference learning (GDD) algorithm which adds a term to the MSPBE objective to constrain how quickly a value function can change. They argue that their approach has a quicker convergence rate, and empirically demonstrate in several examples with linear function approximation that it substantially improves over existing gradient-based TD methods.

### Review

I like the simplicity of the proposed method, and its intuitive interpretation as a value-based trust region. However, I have the following questions and concerns:

1) There doesn't seem to be any information regarding how many independent runs were performed in the empirical evaluation, and there was no no reported statistical significance testing. Can the authors clarify this information, and comment on the significance of the results?

2) While it led to improvements over GTD2, it largely didn't improve over regular (semi-gradient) TD apart from Baird's counterexample, which was designed to make TD fail. As such, I don't think the addition of a new parameter was convincingly justified. Some of the results seemed to suggest that the improvement grew as the state space/complexity increased, that it may be the case that the evaluation falls a bit short on exploring more complex environments. While the breadth of the ablation studies is really nice, we observe similar trends in many neighbouring figures that the space in the main text from showcasing the many different configurations could be summarized with representative examples, and the additional space could have been used to provide some additional experiments/insights (like those suggested in the discussion).

3) From how modular the addition of the term is to the objective, have the authors tried incorporating the regularization to semi-gradient TD? Is there anything about the semi-gradient update that bars its use? TD generally performed really well in the paper's evaluation (outside of Baird's counterexample) that it would make a stronger case if the extension was demonstrated to be more generally applicable, and that it consistently improved over the methods it was applied to. This sort of ties into what was described in 2), where what was presented seems to fall a bit short, and how the space could have showcased a bit more.

4) While the paper's focus was on the case of linear function approximation, can the authors comment on how readily the approach can be extended to the non-linear case? GTD methods have not seen as much adoption as their approximate dynamic programming counterparts when combining TD methods with non-linear function approximation, that it can raise questions as to how the methods scale to more complicated settings.

Given the above, I am erring toward rejection at this time. I think 1) is a rather significant issue that needs to be addressed, and I'm willing to raise my score if that, and my other concerns, can be sufficiently addressed.

----- Post Discussion -----

Taking the other reviews and the authors' response into account, I still maintain my score. While I agree that it's good to be thorough in something clear and simple, it can still be done to a point of redundancy, and consequently seem less thorough in the overall picture and claims made. I'm still largely unsure on the choice to only apply the supposedly modular extension to GTD2, and not try it with TD which seemed like a clearer winner (apart from Baird's counterexample). As others suggested, there are additional methods which might be good to compare to, and other evaluation metrics might make more sense for the claims being made. Many of my concerns were largely brushed off as future work, that little got addressed- without having to carry out the experiments, high level comments/current thoughts could be provided regarding how readily the approach can extend to the scenarios suggested, or if there are nuances that need to be worked out, etc.

---

> ### Author Response · Authors · 2020-11-24
> **the empirical studies are updated in the revised version**
>
> Re to 1: In this revised version, we use 20 runs, and reported the s.d. as the error bar. We have included this information in the updated version of our manuscript. We believe that, with these more careful quantifications, we can, within the context of the tasks that we consider, confidently support the claims that Gradient-DD learning leads to faster convergence than related algorithms, particularly as the size of the state space grows large.
>
> Re to 2: In response to the preceding comment, we have increased the number of runs and included error bars in our numerical experiments. The updated results show more clearly the advantage of our algorithm over TD learning in the Random Walk and Boyan Chain tasks, particularly as the size of the state space increases. Regarding the breadth of the study, we agree that there is a tradeoff between analyzing a few tasks relatively thoroughly versus showcasing the algorithm on a larger variety of examples. We have elected to hew closer to the former approach in order to understand the algorithm relatively comprehensively within a relatively simple context. We are working on a future paper that will apply the idea to a wider array of problems, including nonlinear function approximation with neural networks.
>
> Re to 3: As pointed out by the reviewer and in the second paragraph of our Discussion section, the method that we introduce can indeed be included in a modular way as part of other algorithms for value estimation. We have not yet done this with semi-gradient TD but consider its application there and to other algorithms a promising direction for future work.
>
> Re to 4: As we describe in the Discussion, we agree that this is a critical direction for future research, which we are currently working on. In order to keep the length of the paper under control, we have decided to present the algorithm in the simplest possible context here before scaling the idea up to neural networks in a future paper.

---

### Official Review · AnonReviewer3 · 2020-10-28
**Interesting approach, but needs more work**

**Rating:** 5
**Confidence:** 4

**Review:**

EDIT: After reading the other reviews, the author's responses, and thinking more about the concerns raised, I have increased my score. However, I still recommend rejection because of questions around the hyperparameters used in the experiments.

---

### Summary:
The paper introduces a regularized mean squared projected Bellman error objective function where the regularizer penalizes large changes to the estimated value function. This regularized objective function is used to derive a GTD2-like algorithm where updates to the value function weights are penalized. The paper claims an improved rate of convergence, and empirically investigates the proposed algorithm on tabular random walks, the Boyan Chain environment, and Baird’s counterexample.

### Pros:
+ paper proposes interesting new method
+ paper includes theoretical argument for proposed method
+ paper empirically investigates proposed method

### Cons:
- concerns about soundness of method
- concerns about originality, clarity, and quality

### Decision:
At the present time I recommend rejecting the paper until the following concerns can be addressed.

### Soundness:
Does the proposed modification to the MSPBE change the underlying problem being solved? Is the solution to the regularized MSPBE the same as the solution to the original MSPBE, even with function approximation? The fact that Gradient-DD(4) did not converge on the Boyan chain is very concerning. The motivation for GTD2 is to converge when used off-policy with function approximation. If the proposed modifications lose the convergence guarantee then why not just use conventional TD off-policy?

### Originality:
- There are no references to prior work on convergence rates of GTD2 in section 4. The analysis seems like it was based on an existing analysis, but nothing is cited.
- There is no explicit related work section, which would help clarify the novelty of contributions and would help position the paper within the existing literature.

### Clarity:
- Section 4 (improved convergence rate) is poorly explained, and very difficult to follow.
- Section 5 doesn't mention beta—the step size for the auxiliary weights. Earlier in the paper kappa is referred to as a regularization parameter, but in section 5 it's called a step size parameter and annealed?
- There are several statements that don’t make sense to me:

    - “the regularization term uses the previous value function estimate to avoid large biases in the updating process.”
The use of the word “biases” here is confusing and conflicts with the statistical notion of bias. Updates to weights would generally not be considered “biases” in the statistical sense. However, the regularization term can be thought of as biasing the optimization towards solutions with certain qualities.

    - "[importance sampling] is useful for decreasing the variance of parameter updates"
Using importance sampling to correct the difference between the target and behaviour policies usually increases the variance of parameter updates. IS shrinks updates that occur more often than they would when following the target policy, and enlarges updates that occur less often than they would when following the target policy. The average distance from the mean update can be larger than without importance sampling.

    - "In effect, the regularization term encourages re-experience around the estimate at previous time step, especially when the state space is large."
What does “re-experience” mean?

    - “accelerate the GTD2 algorithm” The word “accelerate” is used several times in the paper to describe the Gradient-DD update, but the idea of penalizing large updates to the value function weights conflicts with the conventional meaning of acceleration in optimization (using past information to make larger changes to weights as is done with Nesterov acceleration, momentum, ADAM, etc.), which is confusing. Penalizing updates to the value function weights would actually slow the changing of the value function weights, not accelerate it. This might allow the second set of weights to learn better estimates of the expected TD error (because the expected TD error is changing as the value function weights change), which could account for the performance increase over GTD2.

### Quality:
- Best performance in the final episode is not an appropriate way to determine the "best-performing" parameter settings when the paper makes claims about the speed of learning of various methods. The parameter settings that result in the lowest error at the end of training will not in general be the parameter settings that result in the fastest learning (i.e., smallest area under the curve). If the paper is going to make claims about learning speed, then the parameter settings should be selected based on the smallest area under the curve. This might be why TDC performs so poorly in these experiments when it out-performs GTD2 in other papers (see Ghiassian et al. 2018; TDC is called GTD in that paper) and intuitively should perform similarly to conventional TD in early learning when the correction weights are near 0. This seems like a serious issue to me; the experiments may need to be re-run with different parameter settings that better match the claims the paper is making about learning speed.

### Suggestions for improvement:
- In addition to addressing the concerns mentioned above, consider adding a related work section that explicitly compares and contrasts the most relevant related methods.
- Consider motivating Gradient-DD more along the lines of TRPO, REPS, and other algorithms that penalize large changes to the weights being learned instead of motivating it as accelerating GTD2.
- Actually, it would be better to do some simple experiments to test why the regularization improves performance over GTD2. Does it result in the second set of weights learning the expected TD error with greater accuracy? Can the same effect be achieved by a two timescale approach where the value function weights are updated with a smaller step size than the second set of weights? If not, it would provide more support for the proposed method.
- Despite the concerns listed in this review, I actually think this paper has a very interesting premise and deserves further study and investigation.

### Misc. details:
- A sentence trails off in the first paragraph of the introduction.
- ”where this term originates from the squared bias term in the objective (6)” Equation 6 seems to be the GTD2 update rules, not the objective function.

### References:
Ghiassian, S., Patterson, A., White, M., Sutton, R. S., & White, A. (2018). Online off-policy prediction. arXiv preprint arXiv:1811.02597.

---

> ### Author Response · Authors · 2020-11-24
> **replies to soundness, originality, clarity, and quality of the method**
>
> <Soundness>:
>
> - Re: Comparing GDD and GTD2, the difference between them is due to V_{n}-V_{n-1}. As the gap between V_{n} and V_{n-1} becomes smaller, less penalty is used here. Thus, intuitively, the convergence of GDD holds as in GTD2.
> Technically, the regularization term makes the update equation a system of ODE equations containing second-order differences, instead of a system of first-order difference equations as in GTD2. This technical issue has prevented us from rigorously proving the convergence of our method, which is why we focus on empirical results in this paper.
> In fact, we have made use of the ODE approach (Borkar and Meyn 2000) to prove the convergence of GTD under constant step sizes. However, in order to apply this approach, we were forced to restrict the step sizes to be constant. This requirement implies that our theoretical result is a weaker convergence than converging with probability 1 based on tapering step size. Empirically, Gradient-DD works well or even better in the case of tapering step size compared with the case of constant step size, suggesting that a stronger convergence proof may be possible. In response to the reviewer’s comments, we have included the weak convergence result, which had been omitted from our earlier submission, in the main text of the updated version of the manuscript, with details provided in the appendix. Although we have not succeeded in doing it so far, proving the stronger convergence with tapered step size is a worthy direction for future theoretical study.
>
>
> <Originality>
>
> - Re: Maei (2011) provides an analysis of the convergence rate similar to ours for the case of the GTD2 algorithm, and we have included a citation to this work in the text. We have added a related work subsection in the Introduction.
>
> <Clarity>
>
> - For the comment of "Section 4 is poorly explained..."
> Re:  We have added several sentences of explanatory text following Eqs. (13) and (14) in Section 4. This includes references to the general approach of analyzing eigenvalues to study the convergence rates of ODEs, as well as to an application of this approach to the GTD2 algorithm.
>
> - For the comment of "Section 5 doesn't mention beta..."
> Re: We apologize for this missing information. In the first version of our submission, we set beta=alpha. In the revision, we also consider the effect of beta by tuning the ratio between beta and alpha. The regularization parameter kappa can be considered a step size if we allow it to be dependent on the time step.  To avoid confusion, we have decided to present the case in which kappa is independent of the time step in the main text (Case 2 in Section 5). The case in which kappa depends on time step has been moved from the main text to the appendix.
>
> - For the comment of some statements.
> Re:  Thanks for pointing out these potential sources of confusion.  They are all changed. We also have avoided this word “accelerate the GTD2 algorithm” in the updated version of the manuscript.
>
> <Quality>
>
> - For the comment of “determine the "best-performing" parameter settings”,
> Re: Because we have compared the final MSEs under different learning algorithms after a fixed number of episodes (rather than after the training has converged or reached some threshold value), we believe that our claims about the speed of learning are justifiable. Unfortunately, there is no perfect metric for this. Comparing areas under the curve requires making a somewhat arbitrary decision about how many episodes to use for computing the area and/or a threshold on the MSE to define when the algorithm has converged, and this method could produce misleading or difficult-to-interpret results if either of the algorithms does not fully converge to the same value, as appears to happen for some of the algorithms that we compare.
>
> <Suggestions for improvement>
>
> - For the comment of adding a related work section,
> Re: Thanks for these suggestions. We have added a section explicitly connecting our work with related work, and have included some discussion on the connection of our work to these algorithms in the newly added section Related Work.
> And the issues in the <misc.details> have been corrected.
>
> - For the comment of beta,
> Re: Thanks for this suggestion. We have added a new figure in which we consider the random walk task with different ratios of beta and alpha. From this new figure we draw two conclusions. First, the performance of our algorithm is relatively robust to the choice of the ratio of these parameters. Second, and more relevant to the reviewer’s comment, the GTD2 baseline algorithm is only modestly enhanced by allowing for beta > alpha, with performance still lagging far behind that of our algorithm. This suggests that the improved performance of our algorithm cannot be straightforwardly achieved simply by applying the two-timescale approach that the reviewer suggested.

---

> > ### Comment · AnonReviewer3 · 2020-11-25
> > **Changes look good, but still concerned about empirical methodology**
> >
> > The newest revision looks much improved, but I'm still concerned about the empirical methodology.
> >
> > > Because we have compared the final MSEs under different learning algorithms after a fixed number of episodes (rather than after the training has converged or reached some threshold value), we believe that our claims about the speed of learning are justifiable.
> >
> > The issue is that the speed of learning of the algorithms may not be representative because the parameters were selected for final performance, and not for learning speed. For TD-based methods there's usually a tradeoff between learning speed and final performance, so the parameters that result in the best final performance usually do not result in the fastest learning speed. If the paper is making claims about the learning speed of each method, the parameters for each method should be chosen as the parameters that result in the fastest learning speed, not final performance.

---

### Decision · Program_Chairs · 2021-01-07
**Final Decision**

**Decision:**

Reject

**Comment:**

This paper introduces an simple but potentially effective off-policy TD algorithm.

Overall, the reviewers felt the work was incomplete and not yet ready for publication. The all recognized the authors made significant updates to the paper, but serious issues remain with the empirical work: studying the impact of the proposed extension on other algorithms, missing baselines (e.g., TDRC), scope of environments limited similar chain-like domains, significant questions about how best parameter settings where chosen for comparison, etc.

This is clearly an interesting direction. If the authors can improve the experiments and better situate their method if the literature (connecting to the lit in off-policy RL about accelerating and improving off-policy TD methods this will become a solid contribution.